# Children’s Particulate Matter Exposure Characterization as Part of the New Hampshire Birth Cohort Study

**DOI:** 10.3390/ijerph182212109

**Published:** 2021-11-18

**Authors:** Jonathan Thornburg, Yuliya Halchenko, Michelle McCombs, Nalyn Siripanichgon, Erin Dowell, Seung-Hyun Cho, Jennifer Egner, Vicki Sayarath, Margaret R. Karagas

**Affiliations:** 1RTI International, Technology-Advancement-Commercialization, 3040 Cornwallis Road, Research Triangle Park, NC 27709, USA; mmccombs@rti.org (M.M.); edowell@rti.org (E.D.); scho@rti.org (S.-H.C.); 2Department of Epidemiology, Geisel School of Medicine at Dartmouth College, Hanover, NH 03756, USA; Yuliya.Halchenko@dartmouth.edu (Y.H.); jennifer.egner@dartmouth.edu (J.E.); vicki.sayarath@dartmouth.edu (V.S.); margaret.r.karagas@dartmouth.edu (M.R.K.); 3Barry Commoner Center for Health and the Environment, Queens College, City University of New York, Queens, NY 11367, USA; nsiripanichgon@qc.cuny.edu

**Keywords:** particulate matter, personal exposure, woodstove smoke, children

## Abstract

As part of the New Hampshire Birth Cohort Study, children 3 to 5 years of age participated in a personal PM_2.5_ exposure study. This paper characterizes the personal PM_2.5_ exposure and protocol compliance measured with a wearable sensor. The MicroPEM™ collected personal continuous and integrated measures of PM_2.5_ exposure and compliance data on 272 children. PM_2.5_, black carbon (BC), and brown carbon tobacco smoke (BrC-ETS) exposure was measured from the filters. We performed a multivariate analysis of woodstove presence and other factors that influenced PM_2.5_, BC, and BrC exposures. We collected valid exposure data from 258 of the 272 participants (95%). Children wore the MicroPEM for an average of 46% of the 72-h period, and over 80% for a 2-day, 1-night period (with sleep hours counted as non-compliance for this study). Elevated PM_2.5_ exposures occurred in the morning, evening, and overnight. Median PM_2.5_, BC, and BrC-ETS concentrations were 8.1 μg/m^3^, 3.6 μg/m^3^, and 2.4 μg/m^3^. The combined BC and BrC-ETS mass comprised 72% of the PM_2.5_. Woodstove presence, hours used per day, and the primary heating source were associated with the children’s PM_2.5_ exposure and air filters were associated with reduced PM_2.5_ concentrations. Our findings suggest that woodstove smoke contributed significantly to this cohort’s PM_2.5_ exposure. The high sample validity and compliance rate demonstrated that the MicroPEM can be worn by young children in epidemiologic studies to measure their PM_2.5_ exposure, inform interventions to reduce the exposures, and improve children’s health.

## 1. Introduction

Seminal epidemiological studies demonstrated children’s exposure to particulate matter (PM) adversely impacts pulmonary development leading to a long-term trajectory of chronic respiratory diseases such as asthma [1,2,3], bronchitis [4], and recurring acute lower respiratory infections [5]. Children’s lungs are not fully functional until approximately 6 years of age when the number of alveoli plateaus and the alveolar epithelium fully develop [6]. Children also have a higher pulmonary surface area per body weight and a higher respiration rate that increases their potential dose of pollutants. Concurrently, a child’s immune system matures to provide immunity phenotypes [7].

To measure or estimate a child’s PM exposure, previous studies relied upon surrogate measures. Frequently employed PM exposure assessment approaches include the nearest government-operated ambient monitor, deployment of study specific stationary indoor or outdoor monitors, and land use regression models. One reason for surrogate measurements is that the burden of personal exposure monitoring on the children or parents historically has been very high. Children, especially young children (e.g., ages 3 to 5 years), cannot physically wear traditional personal monitoring equipment consisting of a separate pump and sampler that could weigh up to 2 kg. Previous attempts to collect pseudo personal level exposure measurements on young children by the parents keeping a lunchbox-sized package near the child collected valid samples, but the representativeness of the data was questionable because parent surveys reported low compliance because of the burden [8]. The development of next generation, low burden PM exposure monitors has enabled children’s exposure characterization [9,10,11]. However, to date, these PM exposure studies recruited children aged 7 years or older. The use of these low burden devices on young children, who are most susceptible to adverse health outcomes from PM exposure, has not been demonstrated.

As part of the New Hampshire Birth Cohort Study (NHBCS), we report the use of wearable, low burden instrumentation to collect PM_2.5_ personal exposure data from children aged between 3 and 5 years of age. Our objectives were to assess the quality of exposure data collected, identify trends in the integrated and continuous PM_2.5_ and chemical species exposure concentration data, and examine child, parent, and residential characteristics that influence their exposure, including use of a woodstove.

## 2. Materials and Methods

### 2.1. Cohort Description

The New Hampshire Birth Cohort Study (NHBCS) is an ongoing, prospective study of rural pregnant women and their children, and is a pediatric cohort for the NIH-funded Environmental influences on Child Health Outcomes (ECHO) program. The primary aim of the NHBCS is to investigate the effects of environmental exposures, including use of wood stoves, on fetal growth and childhood development. A subset of 272 children 3 to 5 years of age participated in a personal monitoring study of PM_2.5_ exposures from March 2017 to February 2020. Participant demographic information, and home heating and tobacco smoke survey data were collected using the NHBCS Early Childhood Environment Checklist and since July 2019 the ECHO Household Exposure to Secondhand Smoke—Childhood questionnaire (Appendix A). The study was reviewed and approved by the Committee for the Protection of Human Subjects at Dartmouth College, and all participants in the study provided written informed consent.

### 2.2. PM_2.5_ Exposure Data Colleciton

Children’s PM_2.5_ exposure was measured with the RTI MicroPEM™ (Research Triangle Park, NC, USA) placed inside an animal-themed front facing halter vest (Appendix A). Study personnel deployed the MicroPEM during a study visit; participant parents were provided with instructions on how to turn off the MicroPEM prior to shipping to the study office after 72 h (3 days) of sample collection. The MicroPEM is a lightweight (~230 g), quiet (~45 dB) particulate matter sampling device designed to be worn by children and adults. The device sampled PM_2.5_ at 0.5 Lpm for real-time nephelometer measurements every 10 s and subsequent collection on a 25 mm, 3 mm pore PTFE filter (Zefon International, Ocala, FL, USA). Embedded temperature and relative humidity sensors automatically corrected the nephelometer concentration [12]. A 3-axis accelerometer measured children’s activity each second to assess wearing compliance [13]. A rolling standard deviation of the accelerometer vector sum greater than 0.007 signifies the child wore the MicroPEM for that 1-min time interval. In this study, the times when the child was asleep were deemed non-compliant. A MicroPEM sample was considered valid for exposure characterization if the sample duration exceeded 22 h with the pump flow within 0.5 ± 0.05 Lpm and battery voltage greater than 4.1 volts. The 22-h validity criterion is a rule of thumb and represents more than 90% of a day. The pump flow and voltage criteria are indicators that the MicroPEM operated properly throughout sample collection.

### 2.3. Filter Analysis Methods

The integrated filter enables gravimetric and chemical analyses that identify exposures to multiple PM species. Gravimetric analysis consisted of triplicate weights pre- and post-sample collection conducted on a Mettler UTM2 digital balance (Toledo, OH, USA) inside a constant temperature (21 °C) and relative humidity (35%) chamber [14]. The method detection limit for this study is 1.2 μg. A mean blank correction factor of 2.2 μg ± 1.3 μg, developed from 13 laboratory blanks, was applied to all gravimetric mass measurements. Black carbon (BC) and combined brown carbon (BrC) and environmental tobacco smoke (ETS) mass were measured using a custom, 6-wavelength integrating sphere transmissometer [15]. The BC and BrC-ETS method detection limits are 0.5 μg for BC and 0.6 μg for BrC-ETS.

### 2.4. Nephelometer Data Procesing

MicroPEM nephelometer data files with valid PM_2.5_ mass concentrations were processed to clean the data. The 10-s nephelometer data were corrected for the nephelometer offset that prevented measurement of negative concentrations and adjusted for baseline shift if necessary [16]. Adjusted concentrations were averaged to a 1-min scale and then normalized such that the integrated average of the 1-min concentrations equaled the corresponding gravimetric concentration. We further stratified the nephelometer data to omit children with a wearing compliance less than 30% because the time-series data should be representative of personal, not stationary, PM_2.5_. We considered minimum thresholds from 60% to 30%, and selected 30% based on the distribution of compliance values measured in this study and the threshold established for similar analyses performed for other cohorts [9,13,17]

### 2.5. Statistical Analysis

A multivariate analysis of the data subset with wearing compliance greater than 30% was performed to identify parent, child, and household factors that influence PM_2.5_, BC, and BrC-ETS exposure. We restricted models for PM_2.5_ and BrC to the non-smoking households to control for potential confounding (Appendix A), whereas BC levels (which are minimally influenced by tobacco smoke) were analyzed with the complete data. We used multivariate linear regression models to explore the association between log transformed PM_2.5_, BC, and BrC levels and indicators of exposure. Separate models were fitted for each outcome of interest and potential source of exposure including presence of a woodstove (yes/no), average number of hours per day of use of a woodstove, whether a woodstove was their primary heating source, and whether they had a household air filter. In addition, sensitivity analyses were run by limiting study population to the households without air filters. We adjusted for factors related to the measurements of interest in either univariate models or fully adjusted models. For models of PM_2.5_ levels, we included BMI at 3 years of age, average number of hours of sleep per night, and season (fall, spring, summer, and winter) as covariates. For BrC levels, we included the child’s age and BMI at 3 years of age, and for BC levels, we included the mother’s enrollment age, and maternal and paternal level of education in the models. Each season: fall, winter, spring (excluding summer as a reference level), as well as the exposures: wood stove (yes /no), woodstove primary heating source (yes/no), a single air filter in use (yes/no), were modeled as binary variables. Ordinal variables such as maternal and paternal education were treated as continuous variables as well as child’s BMI, age, maternal age at enrollment, hours of sleep per night, and exposure variable hours of woodstove use per day on average.

## 3. Results

### 3.1. Exposure Sample Validity and Wearing Compliance

Of the 272 participants, 258 (95%) provided a valid MicroPEM sample for exposure characterization. Invalid samples resulted from battery failure prior to 22 h of data collection (n = 10), the child removed the inlet cap (n = 2), child turned off the MicroPEM (n = 1), and withdrawal from the exposure study (n = 1). The mean sample duration was 74.6 ± 17.3 hrs, with a maximum of 138 hrs. The wearing compliance of the 258 children with a valid sample (Figure 1) was normally distributed with a cohort mean of 46% (SD = 15%, IQR = 18.5%), minimum to maximum range from 10.9% to 82.5%, and six outliers. Compliance values greater than 80% resulted from the valid exposure monitoring period spanning 2 days and 1 night. Wearing compliance was not influenced by parent characteristics such as age, gender, education, or tobacco use, or child gender or BMI, or season (Appendix A).

### 3.2. PM_2.5_ Filter Mass and Carbon Species Concentrations

Of the 258 valid MicroPEM samples, 242 filters had valid gravimetric and transmissometer results. A negative mass after filter blank correction was most common (n = 15) and one filter was punctured during post-collection analysis. The median PM_2.5_ mass concentration was 8.0 μg/m^3^ (GSD = 1.9). The median BrC-ETS concentration was higher than BC (3.5 μg/m^3^ versus 2.3 μg/m^3^) but the GSDs were equal at 2.0 (Figure 2). The fraction of BrC-ETS and BC that comprised the total PM_2.5_ mass were consistently between 0.39–0.47 and 0.28–0.39, respectively (Figure 3). PM_2.5_ mass concentrations lower than 2.5 μg/m^3^ were dominated by BC and BrC-ETS was not detected. The BrC-ETS and BC fractions began to decrease when PM_2.5_ mass concentrations were higher than 34 μg/m^3^.

### 3.3. PM_2.5_ Nephelometer Data

We collected 240 valid nephelometer files from the 272 MicroPEMs deployed. Thirty nephelometer files were deemed invalid because a valid gravimetric correction could not be applied. Two nephelometer files were deemed invalid because of multiple shifts in the concentration baseline of ±15 μg/m^3^ during data collection. Mean 1-min average PM_2.5_ concentration was 6.5 μg/m^3^ with a range from 1.6 to 116.6 μg/m^3^ (Figure 4, Appendix A).

The distribution of PM_2.5_ concentrations by hour of day illustrate when elevated PM_2.5_ exposures occurred for this cohort (Figure 5, Appendix A). The PM_2.5_ concentration time series distributions exhibited higher concentrations in the morning and evening relative to the afternoon and overnight. Eleven of the fourteen highest hourly average PM_2.5_ concentrations were measured between 8 PM and 7 AM and in residences with woodstoves. The other three highest hourly average PM_2.5_ concentrations coincided with typical cooking times (e.g., 6 PM to 8 PM). Hourly PM_2.5_ concentration median and quartiles across the 240 valid nephelometer files follow similar temporal pattern. Although the temporal variability in the median and 1st quartile spans a smaller range, spanning < 4.8 μg/m^3^ versus 8.7 μg/m^3^ for the mean and 3rd quartile, any of the descriptive statistic measures can illustrate temporal trends.

### 3.4. Woodstove Smoke Influences on PM_2.5_ Exposure

We observed statistically significant association between personal PM_2.5_, BrC, and BC concentrations and woodstove smoke exposure, expressed as woodstove use (yes/no) and the average number of hours of woodstove use per day (Table 1). Results of sensitivity analyses were consistent with the primary results (Appendix A). Using a woodstove was associated with 17.4% (95%CI: 3.0–33.9%) increase in personal PM_2.5_ levels. Similarly, each additional hour of woodstove exposure was associated with 1.2% (95%CI: 0.3–2.0%) increase in personal PM_2.5_ levels. Use of an air filter decreased PM_2.5_ levels 23.3% (95%CI: −35.3%, −9.2%). Use of a woodstove was associated with 22.3% (95%CI: 7.1%, 39.6%) increase in personal BrC levels and each additional hour of woodstove exposure was associated with 1.0% (95%CI: 0.1%, 1.8%) increase in BrC exposure. Similarly, having a woodstove was associated with an 18.9% (95%CI: 4.3%, 35.7%) increase in BC personal levels and each additional level of woodstove exposure was associated with a 1.6% (95%CI: 0.7%, 2.5%) increase in BC exposure. Woodstove use as the primary heating source as a dichotomized variable was not clearly associated with PM_2.5_ or BrC levels, but was related to a 13.3% (95%CI: 0.4%, 27.9%) increase in BC.

## 4. Discussion

In a rural cohort from northern New England, young children wore the MicroPEM as instructed. Although the mean compliance of 46% seems low, this value does not credit the child for compliance while asleep with the MicroPEM nearby, as is our typical practice [13]. Parents typically adhered to the 72-h data collection target. Instances with data collection after more than 72 h occurred when parents forgot to stop the MicroPEM, with occasional instances of data collection for more than an extra 24 h. Data collection periods between 22 and 48 h typically resulted from the child refusing to wear the vest with the MicroPEM. Periods shorter than 22 h resulted from battery failure.

The PM_2.5_, BC, and BrC-ETS concentrations and corresponding ratios suggest that combustion sources are likely the primary components of the children’s PM_2.5_ exposure. BrC-ETS and BC comprise between 39 and 44% and between 28 and 33% of the PM_2.5_ mass, respectively, for the middle two-thirds of the measurements. The cumulative BC and BrC-ETS contribution to the PM_2.5_ mass is of 67% to 77% across all samples. This high percentage suggests significant contributions from woodstoves, cigarettes, cooking aerosol, or traffic related air pollution. For the lowest 5% of PM_2.5_ concentrations, BrC-ETS is not present, but the presence of BC is suggestive of diesel engine or candle emissions. Black carbon and ultrafine particle exposure are associated with adverse respiratory health outcomes in children [18,19,20]. At the highest PM_2.5_ concentrations, the BrC-ETS fraction decreases but the BC fraction remains constant potentially indicative of exposure to high concentrations of PM_2.5_ from multiple sources. Although personal and indoor stationary measures are not a direct comparison, our values agree with a previous study of indoor PM_2.5_ concentrations in Northern New England residences with woodstoves, but are lower than other reported indoor PM_2.5_ concentrations in homes with woodstoves that ranged from 17.5 to 32.4 μg/m^3^ measured in homes with woodstoves [21,22,23]. Our median BC and BrC-ETS concentrations were higher than those reported by Fleisch et al. (0.23 μg/m^3^ and 2.80 μg/m^3^, respectively) possibly because of the personal versus indoor stationary measures, and parents that smoked cigarettes were enrolled in our study, even though we restricted the multivariate analysis to non-smoking households [21].

Temporal trends in the mean hourly PM_2.5_ concentration followed a logical pattern also observed in other studies [24,25,26]. Starting at 4 to 5 AM, PM_2.5_ concentrations are at a minimum before increasing to a peak of 13.9 μg/m^3^ as the children and their families conduct their morning activities. A decrease in children’s PM_2.5_ exposure to a mean range from 9.2 to 10.2 μg/m^3^ starting around lunch through late afternoon suggests a quiet time or a cleaner microenvironment possibly outside the home. PM_2.5_ concentrations increase again in the early evening, likely caused by typical evening activities, before becoming stable through 11 PM. PM_2.5_ concentrations, then slowly decay while the family is asleep.

The maximum hourly average PM_2.5_ concentrations occur overnight or in the evening in residences with woodstoves. This strongly suggests these young children that reside in a residence with a woodstove receive their peak exposures overnight from woodsmoke or from cooking. Another woodstove smoke exposure study found hourly mean PM_2.5_ concentrations were most strongly associated with residential activities, where the dominant sources were frying food (34.5 μg/m^3^), wood stove use (26.4 μg/m^3^), fireplace use (25.4 μg/m^3^), and candle burning (20.3 μg/m^3^) [27].

Our analysis supports the hypothesis that woodstove use was a determinant of the children’s PM_2.5_ exposure and the presence of one or more air filters may help mitigate PM_2.5_ exposure. Use of woodstove and hours of use were more clearly associated with PM_2.5_ exposure than use of woodstove as a primary heating source. This may be because woodstove use and duration of use irrespective of whether it is the primary heating source is more predictive or because of imprecision of our estimates for primary heating. Other studies of indoor air quality inside homes that use wood stoves for residential heating also found woodsmoke was an important source of household PM_2.5_. Fleisch et al. (2020) found significantly higher organic carbon and potassium, and elevated, but not statistically significant, differences in PM_2.5_ and BC concentrations in the 29% of homes that used a woodstove compared with those that did not [21]. A Norwegian study that used the MicroPEM as a stationary monitor found mean hourly indoor PM_2.5_ concentrations were higher (*p* = 0.04) for the 14 homes with wood stove use (15.6 μg/m^3^) than for the 22 homes without (12.6 μg/m^3^) [27]. An indoor air cleaner intervention trial found the filtration system reduced PM_2.5_ concentrations from woodstoves by 68% [23,28,29].

## 5. Conclusions

This study is among the first to conduct a personal PM_2.5_ exposure assessment on children during early childhood (i.e., age 3 to 5 years). The exposure assessment techniques used enabled the collection of valid and representative PM_2.5_ exposure data that can be linked to a specific air pollution source. In this case, young children with a woodstove in their non-smoking home have higher PM_2.5_ exposures than children who rely on other sources of heat. A limitation of our study was the inability to adjust for other known factors that influence PM_2.5_ exposure concentration, such as ambient air pollution or environmental inequality, to assess their impact on young children’s exposure. The success of this PM_2.5_ exposure characterization analysis enables future research with other cohorts to associate the direct measurement of a young child’s PM_2.5_ exposure during this susceptible period of their pulmonary development. Our findings suggest that personal monitoring children’s PM_2.5_ exposures from woodstove exposure is a feasible approach for future epidemiologic studies designed to inform interventions to reduce the exposures and improve children’s health.

## Figures and Tables

**Figure 1 ijerph-18-12109-f001:**
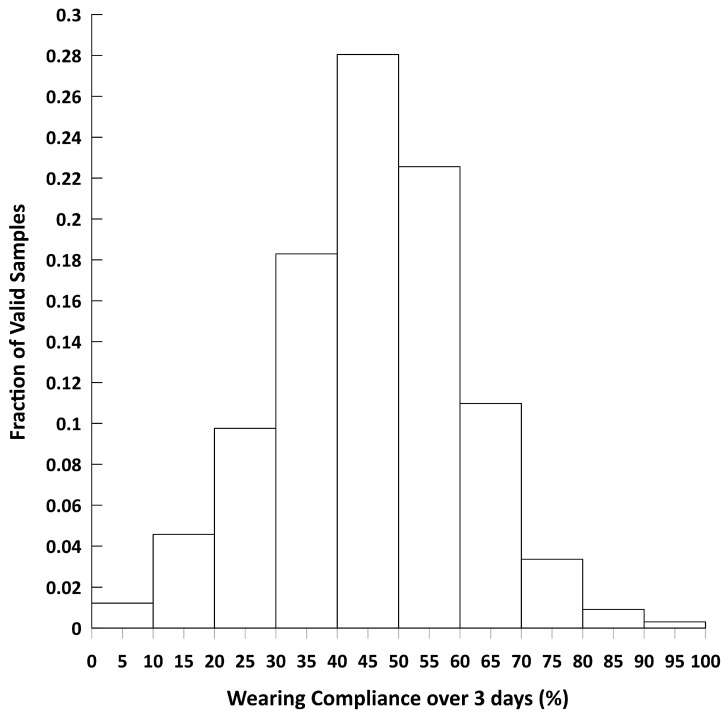
Distribution of wearing compliance from the 258 children with valid MicroPEM samples.

**Figure 2 ijerph-18-12109-f002:**
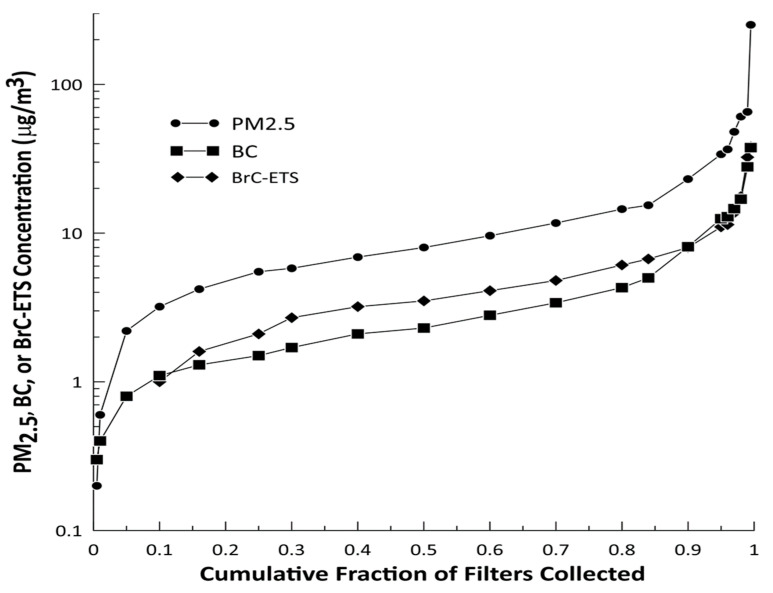
Cumulative distribution of the PM_2.5_, BrC-ETS, and BC mass concentrations measured on the 242 filters with valid gravimetric measurements.

**Figure 3 ijerph-18-12109-f003:**
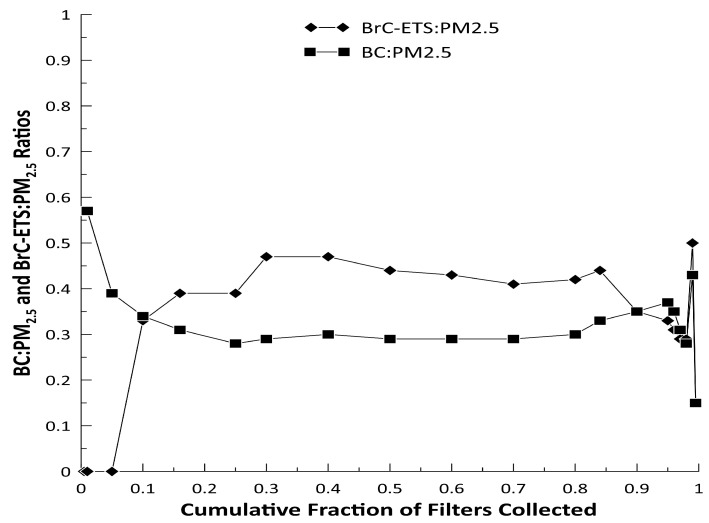
Cumulative distribution of the BrC-ETS:PM_2.5_ and BC: PM_2.5_ mass ratios measured on the 242 valid filter samples. Cumulative fraction calculation based on the PM_2.5_ mass concentration.

**Figure 4 ijerph-18-12109-f004:**
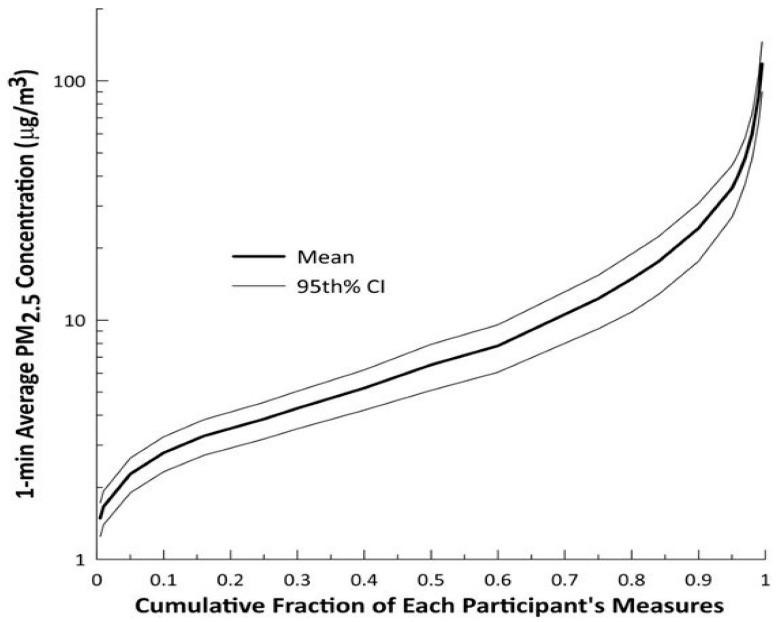
Cumulative distribution of each participant’s 1-min average PM_2.5_ concentrations. The 95th percent confidence intervals illustrate the spread in PM_2.5_ concentrations across each of the 240 valid nephelometer data files.

**Figure 5 ijerph-18-12109-f005:**
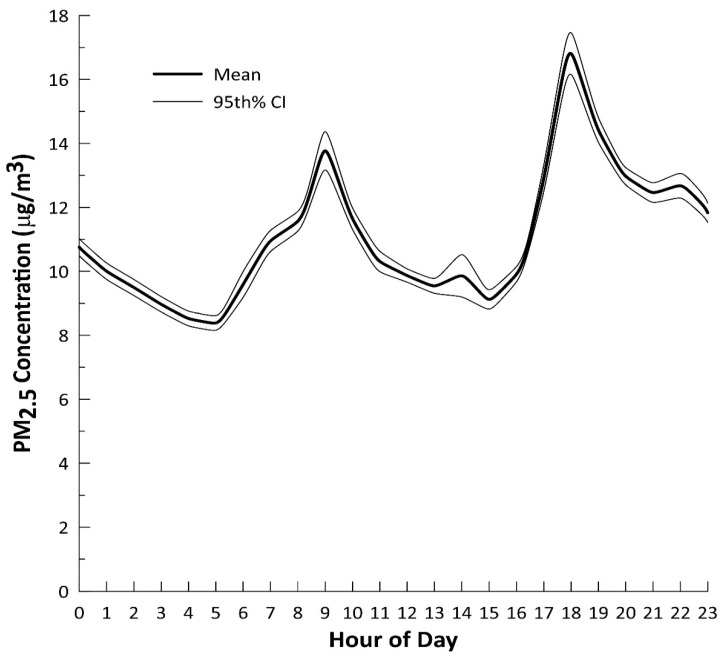
Hourly mean PM_2.5_ concentrations during each hour of the day. The 95th percent confidence intervals illustrate the spread in the hourly PM_2.5_ concentrations across each of the 240 participants with valid nephelometer data files.

**Table 1 ijerph-18-12109-t001:** Percentage change in PM2.5, BC, and BrC levels for each type of exposure for children without ETS exposure.

PM_2.5_ Mass or Carbon Species	Percent Change Per One Unit Change	95% CI	*p*-Value
Total Mass			
Any woodstove use	17.5	(3.1, 33.9)	<0.05
Hours of woodstove use per day on average	1.2	(0.3, 2.0)	<0.05
Woodstove as a primary heating source	2.3	(−8.2, 14.0)	>0.05
Use of an air filter	−23.4	(−35.2, −9.3)	<0.005
BrC			
Any woodstove use	22.3	(7.1, 39.6)	<0.005
Hours of woodstove use per day on average	1.0	(0.1, 1.8)	<0.05
Woodstove as a primary heating source	3.6	(−8.1, 16.9)	>0.05
Use of an air filter	5.2	(−13.3, 27.8)	>0.05
BC			
Any woodstove use	18.9	(4.3, 35.7)	<0.05
Hours of woodstove use per day on average	1.6	(0.7, 2.5)	<0.005
Woodstove as a primary heating source	13.3	(0.4, 27.9)	<0.05
Use of an air filter	−16.4	(−31.2, 1.6)	>0.05

## Data Availability

Processed and original data are available upon request.

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
