# Peer review of "Children’s Particulate Matter Exposure Characterization as Part of the New Hampshire Birth Cohort Study"

_ijerph, 2021, doi:10.3390/ijerph182212109_

Round 1
Reviewer 1 Report
1.The innovation of this study lies in the related research on the evaluation of PM2.5 exposure in early childhood. Its research can correlate the development of young children with the measurement of chronic respiratory diseases (such as asthma).
However, it seems that there is no suggestion and implication for the early development of children after its correlation.
It is suggested that the author put forward a specific explanation at the conclusion section, and put them at the end of the abstract.
2. The author's results only show that wood stove smoke has a significant effect on the children's PM2.5 exposure, and children can wear micropem to measure their PM2.5 exposure.There seems to be no specific suggestions. The author can further explain the contribution and innovation of this paper.
3. Table 1 on p. 6 really doesn't show the author's meaning. Can you explain it further.
Author Response
Thank you for your comments. Your insights will strengthen the paper and caught an obvious error in the manuscript. My responses to each of your comments are below.
- To strengthen the conclusion, we added text to clarify that understanding a young child's PM2.5 exposure can be correlated with distinct phenotypes and thereby reduce the risk of respiratory disease onset and improve disease management. An abbreviated version of this explanation was also added to the abstract
- The innovation of this study was the use of the MicroPEM to collect valid PM2.5 exposure data on children 3 to 5 years of age that identifies a source of their exposure. This point was clarified in the last paragraph of the introduction and in the conclusions.
- Table 1 on page 6 was included by mistake. That table was deleted.
Reviewer 2 Report
The authors in this manuscript describe a study in which particulate matter exposure of young children (3-5 years old) was assessed by using a wearable sensor. This approach was feasible with a high compliance rate and presented results suggest that in this cohort of children woodstove smoke contributed significantly to PM2.5 exposure. There are a number of specific issues that require clarification as follows:
- What was actually measured in this study? Methods (lines 91-94) indicate that the mass of black carbon (BC) and combined brown-carbon (BrC) and environmental tobacco smoke was measured. The latter in particular is abbreviated as BrC-ETS (line 13) but elsewhere in the manuscript other abbreviations are introduced eg BC-Brc-ETS (line 19), BrC (line 188, Table 1 and other Tables) without being defined. Can BrC be measured independently of environmental tobacco smoke?
- Figure 2. BC and BrC-ETS are contributors to total PM5 levels. What is the source of the non BC and BrC-ETS material? Did they investigate associations between the levels of this component and demographics, exposure sources etc as was done for BC and BrC-ETS. If not, why not?
- It is unclear why certain measures were considered valid.
- lines 83-85 “A MicroPEM sample was considered valid for exposure characterization if the sample duration exceeded 22 hours with the pump flow within 0.5 ± 0.05 Lpm and battery voltage greater than 4.1 volts” Why 22 hours? why 4.1 volts?
- Line 95 What is a valid PM2.5 mass concentration?
- Line 90/91 How was the blank correction factor developed from 13 laboratory blanks. Is it simply the average value? What was the SD of these measurement?
- Figure 4 . What was the minimum duration of measurements required so that a 1-min average PM5 concentration could be determined?
- What was the correlation between the gravimetric and nephelometer measures of PM5?
- Table S1. How can a child have a sleep duration of 32.77 hours
- Table S3A (and others). A non-significant p value to four decimal places is meaningless. In addition, what is the difference between a p value of 0.0339 and one of 0.03 or indeed 0.04. Would you interpret the result differently? I would suggest not and it would then be more appropriate in these and other Tables to indicate p values by the usual convention ie P<0.05, P<0.01, P<0.001 etc
- Materials and Methods should be split into subsections
There are some minor issues regarding the English. As examples only (and there may be more.
- lines 17-18 “Highest PM2.5 exposures were in the morning, evening, and overnight” does not make sense as there can be only one “highest”.
- Line 70 “Children’s’”is not a word
- Table 1 line 160 needs to be deleted
- Line 216 “form” should be “from”
Author Response
We appreciate your thorough review that will substantially strengthen the manuscript. Responses to your comments are below. The text was updated accordingly.
- PM2.5, BC, and combined BrC-ETS mass were measured. Our optical transmittance method cannot distinguish BrC from ETS unless survey data or visual inspection of filters confirms there was zero ETS exposure. This study had ETS exposure survey data. "BC-BrC-ETS" was replaced with "cumulative BC and BrC-ETS" to indicate is the sum of both measurements. "BrC" is only used when samples with ETS (identified from surveys) were omitted from the statistical analysis.
- The scope of the grant did not cover full PM2.5 chemical speciation of the samples. Any qualitative assessment of the other PM2.5 species or their sources would be conjecture. The filters are archived and we hope to obtain additional funding to complete the chemical speciation analyses.
- 22 hours is a qualitative value selected because it represents > 90% of a day. The pump flow and battery voltage are MicroPEM operation specifications used to confirm the unit operated properly. Text was added to include these descriptions.
- Yes, the filter blank correction factor was the mean of 13 measurements. The standard deviation was 1.3 ug. The text was updated accordingly.
- At least 22 hours of continuous 1-minute measurements was required to collect a valid sample.
- The nephelometer data normalized with the gravimetric concentration such that the average nephelometer concentration equals the gravimetric. This normalization is described in the Methods and in Section 3.3.
- A majority of the children participated for 3 days. The 32.77 hours spans the 3 days.
- Corrected.
- Done.
- Minor issues noted by the reviewer were fixed.